# ZnO Nanowires on Single-Crystalline Aluminum Film Coupled with an Insulating WO_3_ Interlayer Manifesting Low Threshold SPP Laser Operation

**DOI:** 10.3390/nano10091680

**Published:** 2020-08-27

**Authors:** Aanchal Agarwal, Wei-Yang Tien, Yu-Sheng Huang, Ragini Mishra, Chang-Wei Cheng, Shangjr Gwo, Ming-Yen Lu, Lih-Juann Chen

**Affiliations:** 1Department of Material Science and Engineering, National Tsing Hua University, Hsinchu 30013, Taiwan; agarwal.aanchal@outlook.com (A.A.); young951700@gmail.com (W.-Y.T.); samxgoodguy@hotmail.com (Y.-S.H.); 2Institute of NanoEngineering and MicroSystems, National Tsing Hua University, Hsinchu 30013, Taiwan; argn.mishra30@gmail.com; 3Department of Physics, National Tsing Hua University, Hsinchu 30013, Taiwan; pir820314@gmail.com (C.-W.C.); gwo@phys.nthu.edu.tw (S.G.); 4Frontier Research Center on Fundamental and Applied Sciences of Matters, National Tsing Hua University, Hsinchu 30013, Taiwan

**Keywords:** ZnO nanowires, nanolaser, WO_3_ insulating interlayer, plasmonics, aluminum

## Abstract

ZnO nanowire-based surface plasmon polariton (SPP) nanolasers with metal–insulator–semiconductor hierarchical nanostructures have emerged as potential candidates for integrated photonic applications. In the present study, we demonstrated an SPP nanolaser consisting of ZnO nanowires coupled with a single-crystalline aluminum (Al) film and a WO_3_ dielectric interlayer. High-quality ZnO nanowires were prepared using a vapor phase transport and condensation deposition process via catalyzed growth. Subsequently, prepared ZnO nanowires were transferred onto a single-crystalline Al film grown by molecular beam epitaxy (MBE). Meanwhile, a WO_3_ dielectric interlayer was deposited between the ZnO nanowires and Al film, via e-beam technique, to prevent the optical loss from dominating the metallic region. The metal–oxide–semiconductor (MOS) structured SPP laser, with an optimal WO_3_ insulating layer thickness of 3.6 nm, demonstrated an ultra-low threshold laser operation (lasing threshold of 0.79 MW cm^−2^). This threshold value was nearly eight times lower than that previously reported in similar ZnO/Al_2_O_3_/Al plasmonic lasers, which were ≈2.4 and ≈3 times suppressed compared to the SPP laser, with WO_3_ insulating layer thicknesses of 5 nm and 8 nm, respectively. Such suppression of the lasing threshold is attributed to the WO_3_ insulating layer, which mediated the strong confinement of the optical field in the subwavelength regime.

## 1. Introduction 

The recent advancements in the miniaturization of semiconductor lasers hold the key to emerging technologies, including biosensing [1,2], optical trapping devices [3,4], optical integrated circuits [5,6], photovoltaic devices [7,8], subwavelength imaging [9], on-chip optical communication [10], and computing systems [11]. Traditional semiconductor lasers are severely restricted by the fundamental diffraction law in optics, which limits the size of the optical cavity in orders of the 3D volume to (*λ*/2*n*)^3^, where *λ* is the free space wavelength and *n* is the refractive index of the dielectrics [12,13]. This size restraint causes the shrinkage and scaling down of lasers. In contrast to conventional lasers, nanolasers based on surface plasmons (SPs) could provide a path to achieve a deep sub-diffraction wavelength regime, by forming a nanoscale coherent light source far beyond the diffraction limit [14,15]. Surface plasmons are quantized waves that are bounded at the interface between a metal and a dielectric and that allow the manipulation of light at the nanoscale [16]. The surface plasmon polaritons show great features when excited optically on metal films, such as strong photon–plasmon interactions beyond the diffraction limit, which can lead to enhancement in deep subwavelength regions. Various efforts have been made in the new age of semiconductor lasers, involving small laser volume sizes and low laser thresholds. In recent years, approaches towards diffraction-unlimited plasmonic nanocavity have been demonstrated [17,18,19,20]. This type of nanocavity strongly confines collective electron oscillations at the metal-dielectric interface, because of its remarkable capability to generate intense optical fields and store optical energy into free-electron oscillations in an ultra-compact cavity. The concept of surface plasmon (SP)-based amplification of stimulated emission of radiation (spaser) was first proposed theoretically by Bergman and Stockman [21]. Later, Noginov et al., inspired by this concept, demonstrated the first spaser in 2009, which utilized optical feedback resonators based on the field coupling of gain media to achieve lasing and could produce a strongly localized coherent surface plasmon mode [22]. In addition, a new type of nanolaser was demonstrated, which depended on a nanowire gain medium that generates photons when coupled with the metal form surface plasmons, separated by a dielectric layer forming a Fabry Perot-type surface plasmon polariton cavity [23,24]. Compared with the nanolasers specified above, ZnO exhibits better coupling in ultraviolet regimes [25,26]. Due to its relatively large exciton binding energy of ~60 meV, which is larger than thermal energy ≈ 26 meV, it allows for coupling of ZnO excitons and surface plasmon at room temperature, which is ideal for gain material [27].

Chou et al. unveiled a low-threshold surface plasmon polariton laser by directly placing the ZnO nanowires on an Al surface, and revealed that the threshold strongly depends on the permittivity combination of metal and semiconductor [28]. In line with recent advancements, and to further suppress the threshold of the SPP laser, herein, we synthesized high-quality ZnO nanowires as the gain medium by using a chemical vapor deposition (CVD) method. To demonstrate the low-laser threshold of SPP, an epitaxial Al film was grown through molecular beam epitaxy (MBE), which efficiently reduced the metal scattering, absorption loss, and SPP damping caused by the surface roughness. Further, as-prepared ZnO nanowires were transferred onto a single-crystalline Al film, and a thin WO_3_ dielectric interlayer was inserted between the ZnO nanowires and Al film. The purpose of introducing the dielectric layer between the metal layer and the optical gain medium was to overcome the intrinsic losses of metals and enhance the propagation length of surface plasmon. Moreover, the dielectric layer significantly compensated for the propagation losses and enhanced the lasing performance. An as-fabricated plasmonic nanolaser device, with a WO_3_ dielectric interlayer thickness of 3.6 nm, achieved strong optical confinement and lower threshold (0.79 MW cm^−2^) lasing, which was eight times lower than previously reported in ZnO/Al_2_O_3_/Al plasmonic lasers [29]. For further comparison, we also fabricated the same plasmonic device with different thicknesses (5 nm and 8 nm) of WO_3_ dielectric interlayers. The obtained results reveal that the lasing threshold of the plasmonic device increases with increasing WO_3_ interlayer thickness. Optical simulations further confirm that the plasmonic device with a WO_3_ dielectric interlayer thickness of 3.6 nm has strong confinement in the spacer region.

## 2. Experiment

### 2.1. Synthesis of ZnO Nanowires

The ZnO nanowires were prepared in a horizontal three-zone furnace via a vapor phase transport and condensation process. A mixture of commercially available ZnO (Advanced Chemicals, 200 mesh, 99.999%, Materion Advanced Materials, Milwaukee, WI, USA) and graphite (Alfa Aesar, 200 mesh, 99.999%, metal basis, Thermo Fisher Scientific Chemicals Inc, Ward Hill, MA, USA) powders, with a weight proportion of 3:1, was used as an evaporation source without any further purification. A 10 nm-thick gold film coated silicon (100) substrate was placed at the center in the quartz tube at the downstream side and heated up to 800 °C. In this step, the gold film acted as a catalyst for nanowire growth and was deposited via electron-beam evaporation. The source materials mixture was loaded into an alumina boat, which was positioned at the heating center of zone 1 in the quartz tube reactor and heated up to 1100 °C. Prior to the growth process, the pressure of the furnace was maintained at 1 × 10^−2^ torr, followed by the introduction of the carrier gas mixture of Ar and O_2_, with a volume ratio of 10:1 at 0.16 torr. The samples’ heating and ramping times were set to 1 h and 80 min, respectively. After the process, the samples were cooled down to room temperature in the furnace.

### 2.2. Growth of Epitaxial Aluminum Film

Epitaxial aluminum films were grown on c-sapphire substrates with high-purity (99.9999%) aluminum source by molecular beam epitaxy. A Knudsen cell was used as the evaporator for the growth of the epitaxial aluminum film, to confirm the precise thickness and to ensure that the deposition rate was highly stabilized. Prior to the evaporation, the substrate was heated up for surface cleaning and reconstruction. With the help of liquid nitrogen, the substrate was cooled down to room temperature. During the growth process, a liquid-nitrogen-cooled substrate was used. The evaporation of high purity aluminum source was maintained at a high deposition rate of ≈ 6.67 nm min^−1^. Finally, the film was annealed and kept for a while at room temperature in an ultra-high vacuum chamber. Structural and topographic properties of MBE-grown Al films were characterized by XRD (X-ray Diffraction, Shimadzu D2 Phaser, Kyoto, Japan) and AFM (Atomic Force Microscopy, Bruker- ICON, Billerica, MA, USA).

### 2.3. Measurement System 

The plasmonic lasing measurement was performed using a micro photoluminescence (μ -PL) system. The system was operated with an Andor Technology Shamrock 500 spectrometer coupled with a thermoelectric-cooled CCD camera (Oxford Instruments, Tubney Woods, Abingdon, Oxon, UK). The ZnO nanowires were excited using a 355 nm pulsed laser diode with a repetition rate of 1 kHz and a pulse width of 2 × 10^−9^ s. A working distance of a 50× objective lens with a 0.85 NA (numerical aperture) was used for focusing the incident laser beam. The focused spot size was approximately 5 μm in diameter. Observed light emitted from the plasmonic device was collected using a detector and analyzed through a spectrometer.

### 2.4. Plasmonic Device Fabrication

The plasmonic laser device structure comprised of ZnO nanowires was dropped onto a high-quality single-crystalline aluminum film deposited with a dielectric WO_3_ interlayer. The aluminum film, with a thickness of 80 nm, was grown on a c-sapphire substrate using molecular beam epitaxy (MBE). Hereafter, a dielectric interlayer WO_3_, with a thickness of 3.6 nm, was deposited on a single-crystalline aluminum film through e-beam evaporation, in a vacuum condition greater than 5 × 10^−6^ torr and with a rate of 0.01–0.02 nm/sec. Then, ZnO nanowires were dispersed in the DI water via ultrasonication, and the solution was dropped-cast on the single-crystalline aluminum.

## 3. Results and Discussion

The device of the plasmonic nanolaser was composed of a single-crystalline aluminum film on which ZnO nanowires were placed, separated by a thin dielectric interlayer forming a MOS structure. The schematic of the plasmonic structure is shown in the Appendix A. Due to the smaller ohmic losses, Al (aluminum) was chosen as the plasmonic medium in the UV regime over other metal materials [30,31].

The root-mean-square roughness of single-crystalline epitaxially grown Al was measured through atomic force microscopy in a 20 × 20 μm^2^ area, as shown in Figure 1a. The root-mean-square (RMS) roughness was 0.32 nm. The X-ray diffraction measurement, as shown in Figure 1b, indicates the (111) peak of Al film and (0006) peak of c-sapphire substrate, observed at around 38° and 42°, exhibiting single-crystalline nature.

Figure 2 depicts the morphology and crystallinity of as-grown ZnO nanowires. Figure 2a represents the XRD pattern of ZnO nanowires prepared on a 10 nm-thick Au-coated Si substrate placed at low-temperature zone 3 at 800 °C for 1 h. The growth procedure schematic is shown in the Appendix A. A dominant diffraction peak [002] indicates a high degree of oriented growth with the c-axis vertical to the substrate surface and shows the single-crystalline nature of the ZnO nanowires. Figure 2b shows the low-magnification and high-magnification top-view SEM (Scanning Electron Microscope, Hitachi SU8010, Tokyo, Japan) images of the ZnO nanowires. It can be seen that a high density of ZnO nanowires grew over the entire surface of the Si substrate, and the diameters of the nanowires were about 90–110 nm. Typically, the length of the nanowires ranged from 4 to 6 μm. From the high-magnification SEM image, it is seen that the nanowires had hexagonal facets. Figure 2c,d shows the typical TEM (Transmission Electron Microscope) and HRTEM (High-Resolution Transmission Electron Microscope, Cs-corrected TEM, JEOL ARM 200FTH, Tokyo, Japan) images of the ZnO nanowires. These results indicate that the ZnO nanowires have a fringe spacing of 0.26 nm and match well with the inter-planar spacing of the (002) lattice plane of ZnO. In Figure 2e, in addition to the HRTEM observation, the analysis of the diffraction pattern indicates that the ZnO nanowire grew along the [002] direction, which is consistent with the XRD results. 

Figure 3a shows the measured power-dependent emission spectra of a 4.19 μm-long single-ZnO nanowire placed on an epitaxially grown aluminum film, pumped at RT with a pumping laser wavelength of 355 nm. This lasing emission characteristic was obtained from the micro-PL measurement system (shown in the Appendix A). Below the lasing threshold, a broad spontaneous emission spectrum is observed. This band emission is centered at 384 nm with FWHM ≈ 13 nm and can be seen at a relatively low pump density of ≈1.95 MW cm^−2^. As the pumping power density increases, one oscillation peak appears with an observed lasing peak with FWHM ≈ 1.3 nm. The corresponding light-in-light-out curve for the nanolaser is shown in Figure 3b. From the log-log plot of the light curve, we can obtain the laser threshold at around 2.50 MW cm^−2^.

In order to suppress the large, internal ohmic losses in metals, a dielectric layer was incorporated between the semiconductor and metal, while maintaining the cavity mode volumes. The dielectric spacer layer not only controls the optical confinement but can also isolate ZnO excitons from quenching at the metal surface. The metallic Al films can facilitate the generation of surface plasmons (SPs) and significantly affect the performance of plasmonic nanolasers. To evaluate the laser performance, stronger interaction between plasmons and gain material was required. We then inserted 3.6 nm, 5 nm, and 8 nm-thick WO_3_ dielectric interlayers to observe the ability of optical confinement. The lasing characteristics for the ZnO/WO_3_/Al system, with different thicknesses of the WO_3_ interlayer, are shown in Figure 3c–h. Accordingly, as depicted in Figure 3c, the lasing emission spectra for the ZnO/WO_3_/Al system with 3.6 nm WO_3_ interlayer thickness was measured from 0.48 MW cm^−2^ to 2.06 MW cm^−2^ with the observed linewidth narrowed down from 5 to 1 nm. A non-linear behavior of light-in-light-out (L-L) curve, with the optical lasing threshold of 0.79 MW cm^−2^, was observed in the ZnO/WO_3_/Al system with a 3.6 nm WO_3_ interlayer thickness (Figure 3d). For further optimization, the emission spectra of the ZnO/WO_3_/Al system consisting of a WO_3_ interlayer thickness of 5 nm was measured, with the optical pumping power from 1.19 MW cm^−2^ to 2.56 MW cm^−2^ and the FWHM ≈ 1.3 nm, as shown in Figure 3e. From Figure 3f, the corresponding light-in-light-out (L-L) curve shows that the output intensity drastically increased at threshold power ≈ 1.88 MW cm^−2^. Subsequently, we inserted an 8 nm-thick WO_3_ dielectric interlayer between the ZnO nanowire and Al film and measured the emission spectra for the ZnO/WO_3_/Al system (Figure 3g), with the optical pumping power measured from 1.24 MW cm^−2^ to 5.55 MW cm^−2^ and the FWHM ≈ 1 nm. The corresponding light-in-light-out curve in Figure 3h shows that the optical intensity at the lasing threshold pump density of 2.43 MW cm^−2^ significantly increased. In short, the lasing threshold shows good agreement depending on the loss-compensation system. Of special relevance, the lasing threshold of the ZnO/WO_3_/Al SPP laser with 3.6 nm WO_3_ interlayer thickness was nearly eight times lower than previously reported in similar ZnO/Al_2_O_3_/Al plasmonic lasers (Appendix A). Moreover, the SPP laser operation threshold increased with the increasing thickness of the WO_3_ dielectric interlayer.

The characteristics of the ZnO nanowire supported on high-quality single-crystalline aluminum film with dielectric interlayer were simulated using the finite-difference time-domain (FDTD) solution. Figure 4 shows the calculated cross-sectional view of the energy density distribution of the plasmonic device, which was comprised of ZnO NW placed on 3.6 nm, 5 nm, and 8 nm-thick WO_3_ interlayers over Al film, at wavelengths of 379 nm, where the hexagon cross-sectional side length was set to 60 nm of ZnO NW. It demonstrates that the strong confinement of electromagnetic energy caused an ultra-small space in the WO_3_ (3.6 nm) gap region, because of the continuation of electric field displacement near the WO_3_ dielectric spacer layer.

Previous work has shown that replacing the different dielectric constant based insulating materials can be useful to reduce the lasing threshold [32]. In addition, the reduced surface roughness of the dielectric spacer results in lower scattering losses. Figure 5a–c represents the surface topography of WO_3_ characterized by AFM. The root-mean-square (RMS) of the WO_3_ (3.6 nm) is 0.278 nm, which is considerably smaller than that of WO_3_ (5 nm) and WO_3_ (8 nm), where the RMS values are 0.5 nm for both, resulting in lower optical loss. The height profile analyzed from AFM is shown in the Appendix A. In order to find out the elemental and compositional information of the e-gun deposited WO_3_ sample, X-ray photoelectron spectroscopy (XPS) was explored. The resulting XPS survey spectrum is presented in the Appendix A, and the individual spectra for O (1s) and W (4f) core levels are shown in Figure 5d,e. We found that only tungsten and oxygen are both present in the film. The ratio of W to O is close to 1:3, with a chemical composition of 19.0% and 58.2%, respectively, and the rest is carbon. Ellipsometry measurements were utilized to analyze the dielectric function of WO_3,_ and are shown in the Appendix A. The results indicate that this layer could be useful to reduce the lasing threshold.

Furthermore, the calculated effective indices for the simulated MOS structures, consisting of ZnO nanowires placed on 3.6 nm, 5 nm, and 8 nm-thick dielectric oxide layers on thick epitaxial Al film, are shown in the Appendix A, with the calculation carried out in eigenmode solver in FDTD mode solutions. The increase in effective index implies that the separation among surface plasmon and exciton is smaller, and, to compensate for propagation losses, the coupling of semiconductor exciton and plasmon should be high enough for the energy rate transfer. 

## 4. Conclusions

We demonstrated a plasmonic device consisting of crystalline ZnO nanowires, separated from epitaxially MBE-grown (on sapphire) Al film by a thin WO_3_ dielectric gap layer. This formed a Fabry–Perot type SPP cavity, with an ultra-low lasing threshold down to ~0.79 MWcm^−2^, operating at room temperature. The threshold value reached 2.50 MWcm^−2^ for a plasmonic laser when no dielectric layer was inserted. The suppression of the ultra-low lasing threshold at optimal thickness can be attributed to the insulating layer, which mediated the strong confinement of the optical field in the subwavelength regime, which was eight times lower than previously reported for ZnO/Al and ZnO/Al_2_O_3_/Al plasmonic lasers. The results indicate that replacing the dielectric material will improve the lasing threshold significantly.

## Figures and Tables

**Figure 1 nanomaterials-10-01680-f001:**
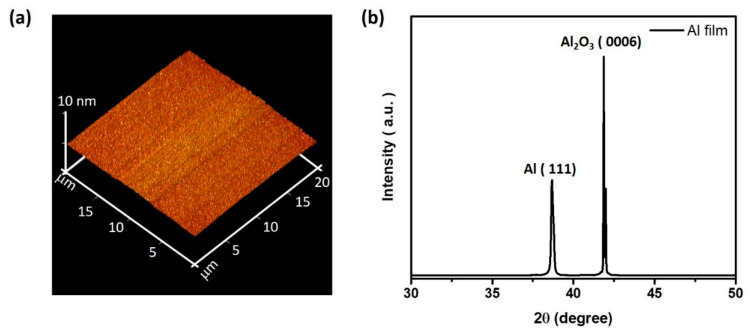
(**a**) 20 × 20 μm^2^ top view atomic force microscope (AFM) image of the single-crystalline Al film. The root-mean-square (RMS) roughness of the surface was about 0.32 nm. (**b**) XRD spectrum of the epitaxially grown Al film with an Al (111) peak and Al_2_O_3_ (0006) c-plane sapphire peak.

**Figure 2 nanomaterials-10-01680-f002:**
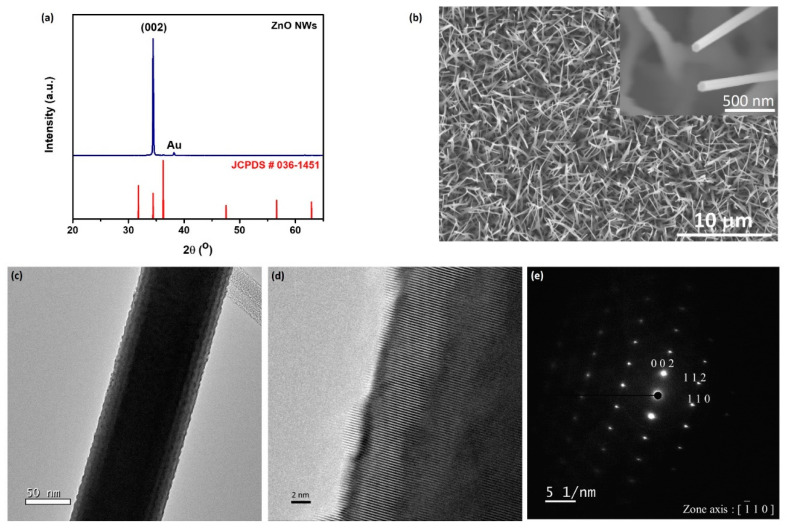
Morphology and crystallinity of the ZnO nanowires. (**a**) XRD spectrum of ZnO nanowires on Si-substrate. (**b**) Low-magnification SEM image of the ZnO nanowires with high-magnification SEM image of ZnO nanowires shown in the inset. (**c**) Low-magnification TEM image of the ZnO nanowire. (**d**) High-resolution TEM image of an individual ZnO nanowire grown along the [002] direction and (**e**) the corresponding SAED (Selected Area Electron Diffraction) pattern.

**Figure 3 nanomaterials-10-01680-f003:**
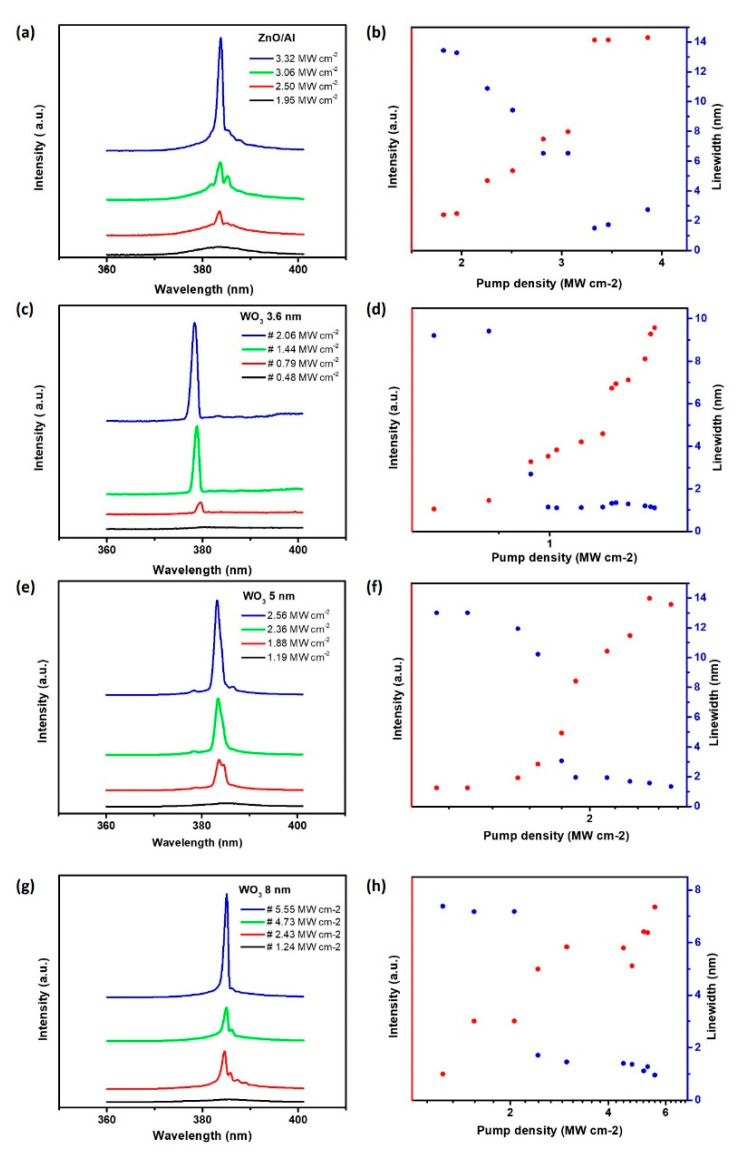
The characteristics of a ZnO nanolaser. (**a**) Measured spectra of the ZnO nanowires on Al film with the pumping power density from 1.95 MW cm^−2^ to 3.32 MW cm^−2^. (**b**) The corresponding light-in-light-out (red dots) curve and linewidth versus input power density (blue dots) at room temperature. (**c**) Measured spectra of the ZnO nanowire directly placed on Al film, with a 3.6 nm WO_3_ layer, and with the pumping power density from 0.48 MW cm^−2^ to 2.06 MW cm^−2^. (**d**) The corresponding light-in-light-out (red dots) curve and linewidth versus input power density (blue dots) at room temperature. (**e**) Measured spectra of the ZnO nanowire directly placed on Al film, with a 5 nm WO_3_ layer, and with the pumping power density from 1.19 MW cm^−2^ to 2.56 MW cm^−2^. (**f**) The corresponding light-in-light-out (red dots) curve and linewidth versus input power density (blue dots) at room temperature. (**g**) Measured spectra of the ZnO nanowire directly placed on Al film, with an 8 nm WO_3_ layer, and with the pumping power density from 1.24 MW cm^−2^ to 5.55 MW cm^−2^. (**h**) The corresponding light-in-light-out (red dots) curve and linewidth versus input power density (blue dots) at room temperature.

**Figure 4 nanomaterials-10-01680-f004:**
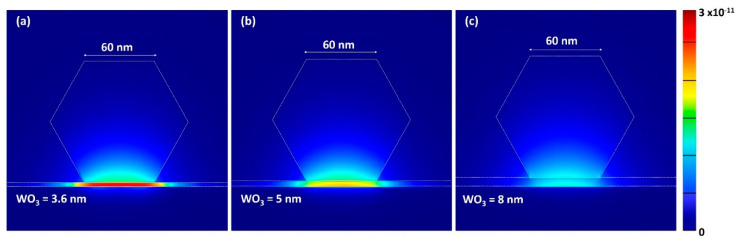
Energy-density distribution of different thicknesses of the WO_3_ spacer layer in ZnO/WO_3_/Al structures simulated by finite-difference time-domain (FDTD) with the excitation wavelength of 379 nm. The cross-section of ZnO nanowire in figures is indicated using white hexagons. (**a**) WO_3_ = 3.6 nm, (**b**) WO_3_ = 5 nm, and (**c**) WO_3_ = 8 nm.

**Figure 5 nanomaterials-10-01680-f005:**
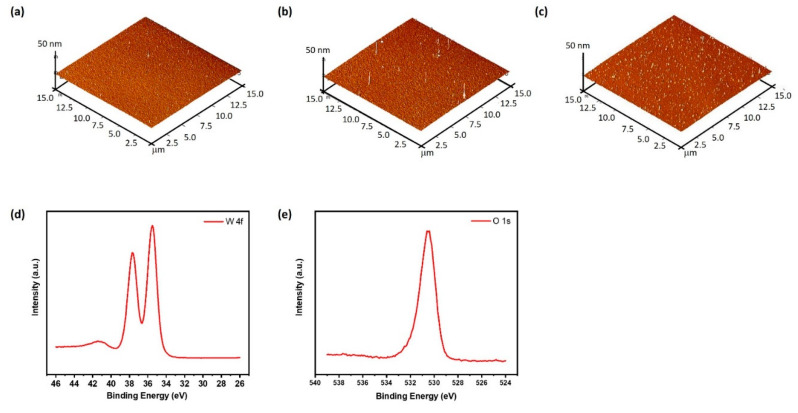
Surface morphologies measured by AFM of different thicknesses of WO_3_ (**a**) 3.6 nm, (**b**) 5 nm, and (**c**) 8 nm. (**d**) XPS spectra of WO_3_ layer at W–4f orbital and (**e**) O–1s orbital.

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
