# Peer review of "ZnO Nanowires on Single-Crystalline Aluminum Film Coupled with an Insulating WO3 Interlayer Manifesting Low Threshold SPP Laser Operation"

_nanomaterials, 2020, doi:10.3390/nano10091680_

Round 1

Reviewer 1 Report

In the paper ‘ZnO Nanowires on Single-Crystalline Aluminum Film Coupled with an Insulating WO3 Interlayer Manifesting Low Threshold SPP Laser Operation’ the plasmonic device consisting of ZnO nanowires, separated from Al film on sapphire by a thin WO3 dielectric gap layer to form a Fabry–Perot type SPP cavities operating at room  temperature is demonstrated. The results indicate that WO3 layer could be appropriately useful to reduce the lasing threshold. The manuscript is well written and clearly presented. It is very useful for scientists in the field of nanophotonics and nanoplasmonics. I recommend this paper be published in NANOMATERIALS after revision.

My comments and suggestions:

  1. Authors should explain the physical reason why the introducing the dielectric layer between the metal layer and optical gain medium is to overcome the difficulty of intrinsic losses of metals and enhancing the propagation length of surface plasmon.
  2. Authors should explain why the transition from Al2O3 to WO3 dielectric interlayer leads to a decrease in the lasing threshold.
  3. The cited paper [15] seems inappropriate to illustrate "...deep sub-diffraction wavelength regime by forming a nanoscale coherent light source far beyond the diffraction limit".
  4. Why WO3 is considered as dielectric media while its dielectric constant is negative in the range from 300 to ~390 nanometers (see Fig. S5(a))?

Author Response

COMMENTS FROM REVIEWER #1

Comment 1:   Authors should explain the physical reason why the introducing the dielectric layer between the metal layer and optical gain medium is to overcome the difficulty of intrinsic losses of metals and enhancing the propagation length of surface plasmon.

Response: We introduced the dielectric layer between metal and optical gain medium because light penetrates on the top surface of the metal and Al metal has small skin depth which causes high propagation losses. The dielectric layer would compensate the propagation losses and enhance the lasing performance. The same has been added in the introduction section of the revised manuscript. (p. 4)

Comment 2:   Authors should explain why the transition from Al2O3 to WO3 dielectric interlayer leads to a decrease in the lasing threshold.

Response:       The k-dielectric constant of WO3 (~15) is higher as compared to that of Al2O3 (~7.8). The high k-dielectric material improves the suppression of charge produced between the metal and the dielectrics which leads to reduction in lasing threshold, hence, we used WO3 dielectric interlayer to replace Al2O3.

Comment 3:   The cited paper [15] seems inappropriate to illustrate "...deep sub-diffraction wavelength regime by forming a nanoscale coherent light source far beyond the diffraction limit".

Response:       The authors appreciate the kind reminder from referee. The cited paper [15] has been updated in revised manuscript. (P. 17)

Comment 4:   Why WO3 is considered as dielectric media while its dielectric constant is negative in the range from 300 to ~390 nanometers (see Fig. S5(a))?

Response: WO3 is used as a dielectric layer in the lasing regime at around 370 nm - 390 nm. We apologies for the misrepresentation of Figure S5(a). If we look closely in the Figure S5(a), dielectric constant of WO3 is positive. In addition, we have updated the Figure S5(a) in revised supplementary information, where we included the inset figure for this wavelength regime, where the dielectric constant of WO3 is positive. The same has been added in Figure R1 in response letter.

Figure R1. Dielectric constants obtained by spectroscopic ellipsometry in the wavelength range of 300-1400 nm for different thickness of WO3 oxide layer. (Please see attached)

Reviewer 2 Report

The manuscript by Agarwal et al. reports experimental results on a plasmonic device consisting of ZnO nanowires, separated from an aluminum (Al) film on sapphire by a thin WO3 dielectric interlayer to form surface-plasmon-polariton (SPP) cavities. The electronic structure of the materials allows the coupling of ZnO excitons and surface plasmons at room temperature, which is ideal for a gain material. The key result of the study is that such a device shows an ultra-low lasing threshold. Its suppression is attributed to the interlayer mediated strong confinement of the optical field.

The different aspects of the nanowire synthesis, the growth of the epitaxial Al film and the plasmonic device fabrication have been described in detail and the performance of the nanolaser has been carefully characterized.

It was a pleasure to read this paper. It is well-written providing many interesting and important details of the plasmonic device. The derived information will certainly help to push the miniaturization of semiconductor lasers even further, which is important for emerging technologies as described in the introduction.

To my opinion the paper is almost ready for publication. The authors might want to improve the quality of some of the figures:

Figure 1(a): larger font size and labels on the axes to be included.

Figure 2: white letters on grey background are difficult to read

Figure 3a, c, e, g: plot the data in the wavelength range of interest (360-400nm). With the present scale spectral features cannot be resolved.

Figure 4: larger font size

Figure 5: larger font size and labels on the axes to be included in a-c

Author Response

COMMENTS FROM REVIEWER #2

Comment 1:   Figure 1(a): larger font size and labels on the axes to be included.

Response:       As suggested, the Figure 1 (a) in the manuscript has been updated. (p. 7)

Comment 2:   Figure 2: white letters on grey background are difficult to read

Response:       As suggested, the Figure 2 has been updated in the revised manuscript accordingly. (p. 8)

Comment 3:   Figure 3a, c, e, g: plot the data in the wavelength range of interest (360-400nm). With the present scale spectral features cannot be resolved.

Response:       Figure 3a, c, e, g have been replotted and revised accordingly. (p. 11)

Comment 4:   Figure 4: larger font size

Response:       As suggested, the font size has been updated in Figure 4 in revised manuscript. (p. 12)

Comment 5:   Figure 5: larger font size and labels on the axes to be included in a-c

Response:       The font size in Figure 5 has been updated in manuscript. (p. 13)
